# Brevisulcenals-A1 and A2, Sulfate Esters of Brevisulcenals, Isolated from the Red Tide Dinoflagellate *Karenia brevisulcata*

**DOI:** 10.3390/toxins13020082

**Published:** 2021-01-22

**Authors:** Masayuki Satake, Raku Irie, Patrick T. Holland, D Tim Harwood, Feng Shi, Yoshiyuki Itoh, Fumiaki Hayashi, Huiping Zhang

**Affiliations:** 1Department of Chemistry, School of Science, The University of Tokyo, Hongo, Bunkyo-ku, Tokyo 113-0033, Japan; irie@yokohama-cu.ac.jp; 2Graduate School of Nanobioscience, Yokohama City University, Seto 22-2, Kanazawa-ku, Yokohama 236-0027, Japan; 3Cawthron Institute, Private Bag 2, Nelson 7010, New Zealand; Patrick.Holland@cawthron.org.nz (P.T.H.); Tim.Harwood@cawthron.org.nz (D.T.H.); FShi@aaapl.com.au (F.S.); 4MS Business Unit, JEOL Ltd., Musashino, Akishima, Tokyo 196-8558, Japan; yoitol@jeol.co.jp; 5NMR Science and Development Division, RIKEN SPring-8 Center, Suehiro-cho, Tsurumi-ku, Yokohama 230-0045, Japan; fumiaki.hayashi@riken.jp (F.H.); huiping.zhang@riken.jp (H.Z.)

**Keywords:** marine polyether, dinoflagellate, red tide incident, harmful algal bloom

## Abstract

Two different types of polycyclic ether toxins, namely brevisulcenals (KBTs) and brevisulcatic acids (BSXs), produced by the red tide dinoflagellate *Karenia brevisulcata*, were the cause of a toxic incident that occurred in New Zealand in 1998. Four major components, KBT-F, -G, -H, and -I, shown to be cytotoxic and lethal in mice, were isolated from cultured *K. brevisulcata* cells, and their structures were elucidated by spectroscopic analyses. New analogues, brevisulcenal-A1 (KBT-A1) and brevisulcenal-A2 (KBT-A2), toxins of higher polarity than that of known KBTs, were isolated from neutral lipophilic extracts of bulk dinoflagellate culture extracts. The structures of KBT-A1 and KBT-A2 were elucidated as sulfated analogues of KBT-F and KBT-G, respectively, by NMR and matrix-assisted laser desorption/ionization tandem mass spectrometry (MALDI TOF/TOF), and by comparison with the spectra of KBT-F and KBT-G. The cytotoxicities of the sulfate analogues were lower than those of KBT-F and KBT-G.

## 1. Introduction

A widespread bloom of the dinoflagellate *Karenia brevisulcata* [1] occurred in the central and south-east coast of the North Island of New Zealand in early 1998 with deadly and devastating consequences to fish and other marine organisms in Wellington Harbour [2,3]. In addition to the devastating damages to marine animals, more than 500 patients were reported during the red tide incident. Characteristic symptoms were respiratory distress, inflammation of skin and eyes, severe headaches, and facial sunburn sensations, which were similar to symptoms caused by *Karenia brevis* [4,5,6]. Further toxicological studies are needed to elucidate the correlation between human illness and the toxins of *K. brevisulcata. Karenia brevisulcata* is morphologically very similar to *Karenia brevis* and *K. mikimotoi*, which produce ladder-frame polyether toxins, brevetoxins [7,8], and gymnocins, respectively [9,10]; however, the cells of *K. brevisulcata* are smaller, and brevetoxins and gymnocins were not detected in the cell extracts. Therefore, the toxins produced by *K. brevisulcata* that caused the toxic event in New Zealand in 1998 were presumably unknown.

Cell extracts of *K. brevisulcata* exhibited potent mouse lethality and cytotoxicity. The extracts were partitioned between chloroform and aqueous methanol under neutral conditions during initial investigations. Two different types of toxins were detected in the extracts [3]. The aqueous methanol layer contained brevisulcatic acids (BSXs), brevetoxin-like ladder-frame polyether compounds possessing side-chain carboxylic acids. BSXs displayed potent cytotoxicity against neuro2A cells with veratridine and ouabain [11,12]. The lipophilic layer contained brevisulcenals (KBTs), which exhibited potent mouse lethality and cytotoxicity against the P388 (murine leukemia) cell line.

NMR and matrix-assisted laser desorption/ionization (MALDI) tandem mass spectrometry (MS) with high-energy collision-induced dissociation (HE-CID) led to the elucidation of the four KBT structures, brevisulcenal-F (KBT-F), KBT-G, KBT-H, and KBT-I, as shown in Figure 1 [13,14]. KBT structural features include molecular weights above 2000 and extensive ladder-frame polyether compounds comprising 24 ether rings, including a dihydrofuran, decorated with hydroxy groups, methyl groups, and conjugated unsaturated aldehyde side chains. The longest contiguous ether ring chain of KBTs comprises 17 units (A–Q). Apart from a single oxidation, the skeletal structure of KBT-F and KBT-H is the same, as is that of KBT-G and KBT-I. KBT-H and KBT-I possess branched primary alcohols on C-2, generated by oxidation of olefinic methyls in the side chains of KBT-F and KBT-G, respectively. Such branched oxidized structures are very rare among polycyclic ethers produced by dinoflagellates. The cytotoxicity of KBT-H and KBT-I was more potent than that of KBT-F and KBT-G, indicating that oxidation of terminal olefinic methyls enhances toxicity. The long contiguous ether ring assembly with an unsaturated aldehyde terminus is analogous to the structures of gymnocins A and B [9,10], but KBTs displayed more potent mouse lethality and cytotoxicity. Structural elucidation of KBT analogues is important for the development and improvement of detection methods [15] and for investigating the mode of action and structure-activity relationships of these toxins, which may be beneficial for developing preventative strategies to mitigate the effects of KBT red tide events, should they occur again. Our ongoing efforts in this area led to the isolation of new KBT analogues, brevisulcenal-A1 (KBT-A1), and brevisulcenal-A2 (KBT-A2), which are sulfate esters of KBT-F and KBT-G, respectively. KBT-A1 and KBT-A2 eluted before KBT-F and KBT-G when using reversed-phase chromatography. In this study, we elucidated the structures of KBT-A1 and KBT-A2 by detailed analyses of their NMR and spiral MALDI TOF/TOF spectra.

## 2. Results

### 2.1. Extraction and Isolation of Brevisulcenals

Brevisulcenals (KBTs) were extracted from mature bulk cultures of *Karenia brevisulcata* employing resin-based isolation. Cells were lysed with acetone and after stirring for 1 h, the cultures were diluted with water and passed through HP20 resin. Brevisulcenals were recovered by washing the resin with acetone. The lyophilized extract was dissolved in MeOH and diluted to 55% MeOH with pH 7.2 phosphate buffer, and then partitioned with chloroform (CHCl_3_). The CHCl_3_ extract was chromatographed on a diol-silica column by stepwise elution with ethyl acetate (EtOAc) and MeOH, EtOAc:MeOH (9:1, 8:2, 7:3, 6:4), and fraction collection was guided by the cytotoxicity assay against mouse leukemia P388 cells and UV absorption. Further purification of KBTs was performed using reversed phase chromatography with a linear gradient elution from 80% to 100% MeOH. The final purification on a reversed phase column with isocratic elution of 80% MeOH led to the isolation of 0.2 mg of KBT-A2 from 150 L of K. *brevisulcata* culture. Isolation and accumulation of KBT-A2 was difficult because of its low abundance and short retention time on the reversed phase column. From 690 L of ^13^C labeled cultures, 0.8 mg of KBT-A2 was generated for NMR studies. On the other hand, only 0.2 mg of KBT-A1 was isolated from non-labeled cultures due to its early elution on reversed-phase columns.

### 2.2. Structural Elucidation of Brevisulcenal-A2

In the positive ion MALDI mass spectrum of KBT-A2, high-intensity signals were observed at *m/z* 2207.8 and 2105.5, and the difference between the two ions was 102 Da, corresponding to desulfonation. Therefore, these ions were identified as a sodium adduct ion [M+Na]^+^ and a desulfonated ion [M−SO_3_Na+H+Na]^+^, respectively. A signal for [M−Na]^−^ ion at *m/z* 2161.7 in the negative-ion MALDI mass spectrum confirmed that KBT-A2 possessed a sulfate ester as a sodium salt. Accurate mass measurement using MALDI-Spiral TOF (Appendix A) [16] indicated that the molecular formula of KBT-A2 was C_108_H_161_O_42_SNa (sodium salt) ([M+Na]^+^ 2207.9971, calcd. 2207.9973). The molecular formula indicated that KBT-A2 was a sulfate ester of KBT-G (C_108_H_162_O_39_).

The UV maximum of KBT-A2 was observed at 227 nm (ε 1.2 × 10^4^). The UV absorption suggested that KBT-A2 has an enal side chain analogous with KBT-G. The ^1^H NMR spectra of KBT-A2 (Appendix A), obtained at 800 and 500 MHz, resembled those of KBT-G. Comparison of the KBT-G and KBT-A2 HSQC spectra (Appendix A) revealed that the ^1^H and ^13^C chemical shifts of KBT-A2 arising from CH-1 to CH-59 were close to those of KBT-G (Table 1).

However, the ^1^H and ^13^C chemical shifts corresponding to the CH-71 region differed significantly. The Δδ_H_ (δ_H-KBTG_−δ_H-KBTA2_) and Δδ_C_ (δ_C-KBTG_−δ_C-KBTA2_) values in the vicinity of CH-71 are summarized in Table 2. The ^1^H chemical shift of H-71 shifted downfield from δ_H_ 5.05 in KBT-G to δ_H_ 5.70 in KBT-A2, thus Δδ_H_ of H-71 was −0.65. Similarly, the ^13^C chemical shift of C-71 shifted downfield from δ_C_ 67.1 in KBT-G to δ_C_ 73.3 in KBT-A2, thus Δδ_C_ of C-71 was −6.2. The Δδ_H_ and Δδ_C_ of CH-71 were greater than those of the remaining protons and carbons in that region. In addition, in the ^1^H NMR spectrum of KBT-G, proton coupling between H-71 and a hydroxy proton was observed, while this was not the case for KBT-A2, due to the substitution of OH with OSO_3_Na. Therefore, it was deduced that the sulfate ester was bonded to C-71 (Figure 1).

As MALDI tandem MS with HE-CID using a MALDI-SpiralTOF-TOF [17] instrument proved useful for the structural determination of KBTs, structural confirmation of KBT-A2 was also conducted with this experimental setup. The sulfate ester in KBT-A2 is a suitable charge site for negative ion tandem MS measurements. A [M−Na]^−^ ion at *m/z* 2162 was selected as the precursor ion for the HE-CID tandem MS experiments. Product ions generated by bond cleavage from both the aldehyde (C-1) and the methyl (C-95) terminals were observed because the position of the charge site, the sulfate ester, resided in the middle of the molecule. Product ion assignments are explained in Figure 2. A product ion at *m/z* 2078 was generated by cleavage of a 2-methylbut-2-enal side-chain (Figure 2a). Product ions arising from ring A to ring Q were clearly observed (Figure 2a). The ions observed at *m/z* 1940, 1870, 1800, 1714, 1658, 1602, 1532, and 1462 confirmed the presence of contiguous six-membered ether rings, C–I, methyl positions on B/C, C/D, D/E, G/H, and H/I junctions, and hydroxy-substitution on ring E (Figure 2a). Prominent product ions at *m/z* 1191, 1121, 1051, and 965 were observed in the spectrum of KBT-A2, and the mass differences between those product ions were 70, 70, and 86 Da, respectively. The presence of these ions strongly supported that rings M to P comprised a 7-membered ether ring, a 6-membered ether ring with a methyl, a 7-membered ether ring with a hydroxy group, and a 6-membered ether ring with a methyl, respectively, analogous to KBT-G. The product ions at *m/z* 895, 825, 689, and 617 confirmed the absence of a hydroxy group at C60 on ring Q, the presence of a 1,2,3-trihydroxypropyl linker, a 6-membered ether ring with two hydroxy groups (ring R), and the location of the sulfate ester (Figure 2a). Below *m/z* 600, only sulfate-related ions at *m/z* 96.8 and 79.8 were observed. Product ions generated by cleavage from the C-95 terminus are assigned in Figure 2b. The product ions at *m/z* 2088 and 2020 were generated by cleavage of the C93–C95 side chain and 2,5-dihydrofuran, respectively. The product ions at *m/z* 1948, 1892, 1836, 1780, and 1710 confirmed the ring arrangement from S to W of KBT-A2 as 6/6/6/6/6/ where the NMR signals overlapped, and the position of the sulfate ester. Thus, these product ions supported the structure of KBT-A2 deduced from NMR spectral analyses.

### 2.3. Structural Elucidation of Brevisulcenal-A1

The UV absorption of brevisulcenal-A1 (KBT-A1) was at 225 nm (ε 0.6 × 10^4^), indicating that KBT-A1 had an enal side chain. MALDI-SpiralTOF (Appendix A) confirmed that the molecular formula of KBT-A1 was C_107_H_159_O_41_SNa (sodium salt) ([M+Na]^+^ 2177.9867, calcd. 2177.9867). This indicated that KBT-A1 is a sulfate ester of KBT-F. Although the very small amount (0.2 mg) of KBT-A1 isolated from non-labeled cultures made it difficult to acquire NMR spectra (Appendix A), the HSQC spectra (Appendix A) of KBT-A1 enabled us to assign the ^1^H and ^13^C chemical shifts by comparison to the HSQC spectra of KBT-F (Table 1). The proton chemical shift for H-71 in KBT-A1 was observed at δ_H_ 5.72, while that of KBT-F was observed at δ_H_ 5.13. The Δδ_H_ value of −0.59 suggested that a sulfate ester resided on C-71, analogous to KBT-A2.

Structural validation of KBT-A1 was also accomplished by a MALDI-SpiralTOF-TOF experiment (see Appendix A). A [M−Na]^−^ ion at *m/z* 2132 was selected as the precursor ion for HE-CID tandem MS experiments. The product ions from ring A to ring Q were clearly observed (Appendix A). Similar to KBT-A2, the product ions observed at *m/z* 1909, 1839, 1769, 1684, 1628, 1572, 1502, 1432, 1331, 1261, and 1161 confirmed the existence of the contiguous six-membered rings C–I and the J–K ring sequence. These product ions were smaller by 30 Da than the corresponding ions of KBT-A2. Prominent product ions at *m/z* 1091, 1035, and 965 were observed in the spectrum of KBT-A1. The mass differences between the product ions were 56 and 70 Da, indicating that rings N and O comprised a 6- and 7-membered ether ring, respectively. The ions at *m/z* 895, 823, 809, 765, and 735 confirmed the presence of a hydroxy group at C60, and the presence of a 2,3-dihydroxypropyl linker. The product ions generated by cleavage of rings S-W, 2,5-dihydrofuran, and the C93–C95 side chain were clearly observed at *m/z* 2058, 1917, 1861, 1805, and 1749 (Appendix A). Therefore, the structure of KBT-A1 is shown in Figure 1.

## 3. Discussion

The cytotoxicity of KBT-A1 and KBT-A2 against mouse leukemia P388 cells was evaluated, obtaining LC_50_s of 10.9 and 3.5 nM, respectively. That of KBT-F and KBT-G was 2.7 and 0.7 nM, respectively This indicated that the presence of a sulfate ester on brevisulcenal reduces its cytotoxicity. KBTs possess rigid polyether assemblies linked by flexible linear alkyl chains and comprise hydrophobic (rings A–Q) and hydrophilic (linkers and rings R–X) portions. The sulfate ester is positioned on the linker in the hydrophilic portion, increasing its polarity. Structural modifications, such as oxidation of the terminal olefinic methyls and substitution by a sulfate ester, influence the activity of KBTs. Marine polycyclic ethers, maitotoxin (MTX) [18], and yessotoxin (YTX) [19] possess two sulfate esters in their framework and show potent mouse lethality and cytotoxicity. Contrary to KBTs, the cytotoxicity of desulfonated derivatives of MTX and YTX are lower than those of MTX and YTX [20,21,22]. Therefore, sulfate esters of marine polycyclic ether toxins play an important role in their biological activities, as they affect target protein interactions.

## 4. Materials and Methods

### 4.1. General Methods

All purchased solvents (FUJIFILM Wako Pure Chemicals, Osaka, Japan) were of the highest commercial grade and used as received, unless otherwise noted. UV-visible absorption spectra were collected on a JASCO V-550 UV spectrometer (JASCO Co., Tokyo, Japan). A JASCO PU-98 pump and a JASCO UV-970 UV detector were used for liquid chromatography. NMR spectra were recorded on three NMR instruments (Bruker CO., Bremen, Germany and JEOL, Tokyo, Japan): At 800 MHz (200 MHz for ^13^C) or 500 MHz. Chemical shift values are reported in ppm (δ) referenced to internal signals of residual protons (^1^H NMR; C_5_HD_4_N (7.21); ^13^C NMR, C_5_D_5_N (125.8)).

### 4.2. Culture Growth and Harvesting

*Karenia brevisulcata* (CAWD82) was collected from the Wellington Harbor in 1998 and was kept at Cawthron Institute Culture Collection of Microalgae (CICCM), Cawthron Institute, Nelson. Bulk cultures (150–250 L batches) were grown in 12 L carboys using 100% GP + Se media under a 12/12 h day/night-timed cool white fluorescent lighting regime, with 25 min of aeration at 30 min intervals. Starter culture (14–21 days old) was added to 100% GP + Se media at a ratio from 1:10 to 1:15. Cultures were maintained for up to 21 days. Aliquots of culture were assessed for cell numbers by inverted microscopy. For ^13^C-enrichment, cultures were augmented at 0 and 10 days with NaH^13^CO_3_ (0.25 g per 12 L). Toxin production was assessed by liquid chromatography-mass spectrometry (LC-MS), following SPE using a 50 mL aliquot of culture extracted with Strata-X (60 mg, Phenomenex Inc., CA), washing with Milli-Q water and 20% methanol, and eluting with methanol or methanol followed by acetone (3 mL each).

Toxins were extracted from mature cultures using Diaion HP20 resin. Briefly, the pre-washed resin was packed in a polypropylene column. *K. brevisulcata* cultures were transferred to a 200 L barrel, and cells were lysed by the addition of acetone to 7% *v/v*. The cultures were allowed to settle for 1 h and then diluted with reversed osmosis purified water (RO water) to 5% *v/v* acetone before pumping at 0.3 L/min first through a filter system then through a HP20 resin column. The column was then washed with water, and the HP20 resin was transferred to a 2 L flask. Toxins were recovered by soaking the resin in AR acetone (1 L) and decanting (×3). The combined acetone extracts were evaporated in vacuo to produce a dried crude extract.

### 4.3. Isolation of KBTs

The crude HP20 extract was dissolved in methanol and diluted to 55% *v/v* with pH 7.2 phosphate buffer. The solution was partitioned with CHCl_3_ (×2), and the combined chloroform fraction containing neutral toxins was evaporated. Brevisulcenals were isolated from the neutral fractions of 1450 L of bulk cultures by column chromatography using a diol cartridge (500 mg) with stepwise elution of ethyl acetate (EtOAc), EtOAc:MeOH (9:1 and 6:4), and MeOH, and guided by the P388 cytotoxicity assay. The EtOAc:MeOH (6:4) and MeOH fractions were combined. Further purification was conducted by preparative HPLC (250 mm × 4.6 mm i.d. Develosil C30-UG-5, Nomura Chemical Co., Japan) with linear gradient elution from 80% MeOH/H_2_O to 100% MeOH and guided by UV absorbance at 230 nm. Final chromatographic purification was accomplished on a Develosil C30-UG-5 column with isocratic elution of 80% MeOH at 1 mL/min. The retention time of KBT-A1 and KBT-A2 was 9–10 min.

### 4.4. MALDI MS and MALDI Tandem MS Measurements

MALDI mass spectra were recorded on a JEOL JMS-S3000 SpiralTOF instrument (JEOL, Tokyo, Japan) using α-cyano-p-hydroxycinnamic acid (CHCA, FUJIFILM Wako Pure Chemicals, Osaka, Japan) as a matrix for the positive ion mode. KBT-A1 and KBT-A2 were dissolved in MeOH:CHCl_3_, mixed with a norharmane matrix, and subjected to MALDI tandem MS measurements in negative ion mode. The product ion mass spectra were recorded with laser irradiation at 349 nm, a frequency of 250 Hz, and acceleration voltage of −20 kV in the first TOF stage. The collision energy was set at 20 keV to afford the HE-CID. Product ions formed by collision-induced dissociation were accelerated by 9 kV for analysis in the second TOFMS stage.

### 4.5. Chemical Properties of Brevisulcenal-A2

Brevisulcenal-A1: Isolated as a colorless amorphous solid: UV maxima (λ) 225 (ε 6000) nm. The high-resolution MALDI-TOF mass spectrum gave [M+Na]^+^ at *m/z* 2177.9867 for C_107_H_159_O_41_SNa_2_ ([M+Na]^+^ calcd. 2177.9867). ^1^H and ^13^C NMR data are presented in Table 1.

Brevisulcenal-A2: Isolated as a colorless amorphous solid: UV maxima (λ) 227 (ε 12,000) nm. The high-resolution MALDI-TOF mass spectrum gave [M+Na]^+^ at *m/z* 2207.9971 for C_108_H_161_O_42_SNa_2_ ([M+Na]^+^, calcd. 2207.9973). ^1^H and ^13^C NMR data are presented in Table 1.

### 4.6. Cytotoxicity

Mouse leukemia cells, P388, were cultured in RPMI 1640 supplemented with 5% penicillin/streptomycin solution and 10% inactivated fetal bovine serum at 37 °C under an atmosphere of 5% CO_2_ in a CO_2_ incubator. Each well of a 96-well microplate was filled with 100 μL of P388 cell suspension containing 1.0 × 10^4^ cells/mL, followed by the addition of 100 μL of KBT solution dissolved in RPMI 1640 medium. The plate was incubated in a CO_2_ incubator at 37 °C for 72 h. After 72 h of incubation, the plate was incubated for another 4 h with 50 μL of 1.0 mg/mL aqueous MTT solution under the same conditions. The obtained precipitate was dissolved in DMSO, and absorbance at 570 nm was measured with a multiwavelength spectrometer. Cytotoxicity against P388 cells was determined using the MTS colorimetric reaction method (CellTiter 96^®^AQueous One Solution Reagent, detected at 490 nm).

## Figures and Tables

**Figure 1 toxins-13-00082-f001:**
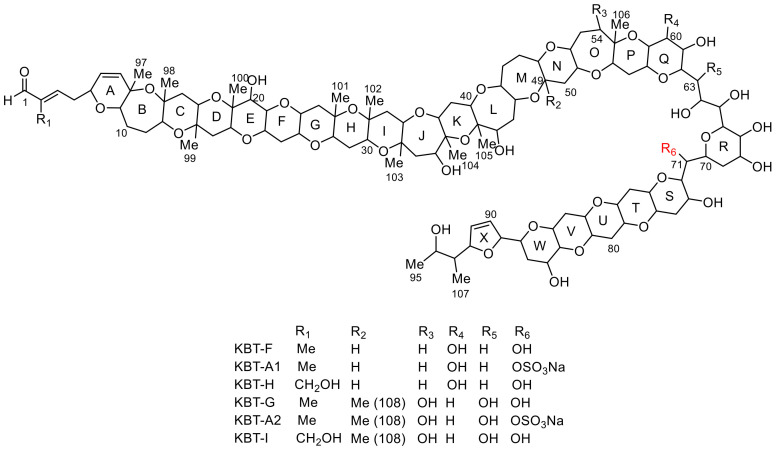
Structures of brevisulcenals.

**Figure 2 toxins-13-00082-f002:**
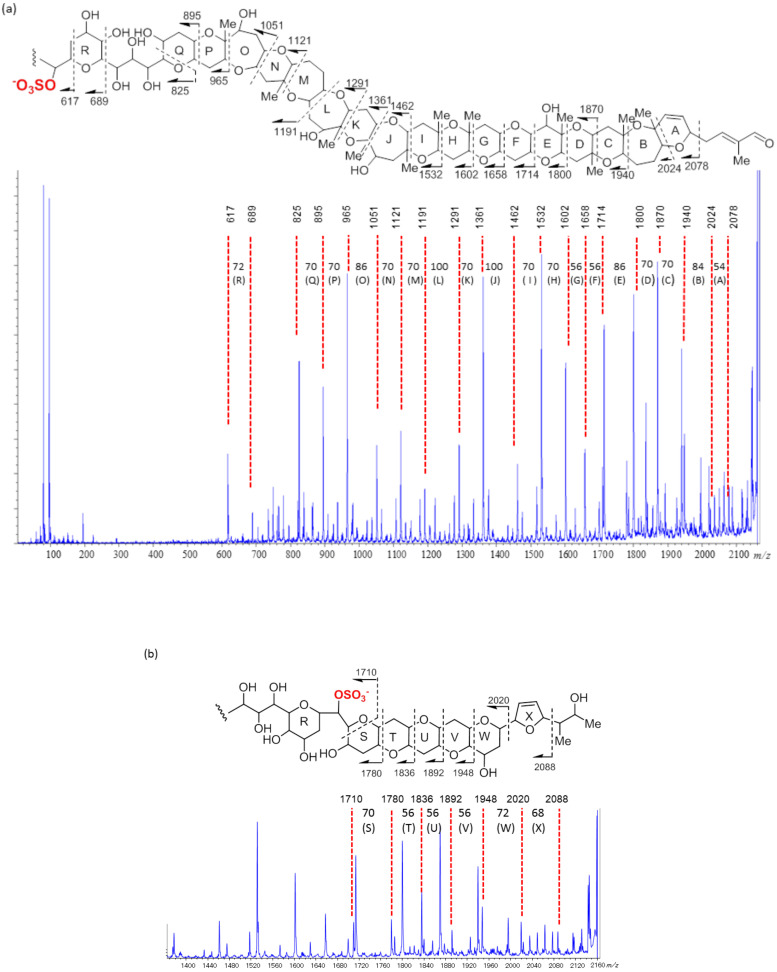
TOF-TOF spectra of brevisulcenal-A2 and assignments of prominent product ions in the partial structures: (**a**) Whole spectrum and assignments from the C-1 terminus, (**b**) expanded spectrum *m/z* 1400–2000 and assignments from the C-95 terminus.

**Table 1 toxins-13-00082-t001:** NMR spectroscopic data (800 MHz, pyridine-*d*_5_) for brevisulcenal-A1 (KBT-A1) and KBT-A2.

Pos.	KBT-A1^a^		KBT-A2^a^		Pos.	KBT-A1^a^		KBT-A2^a^		Pos.	KBT-A1^a^		KBT-A2^a^	
	δ_H_	δ_C_	δ_H_	δ_C_		δ_H_	δ_C_	δ_H_	δ_C_		δ_H_	δ_C_	δ_H_	δ_C_
1	9.51	197.1	9.54	197.1	36	4.11	78.7	4.11	78.3	70	4.92	70.7	4.92	72.6
2		143.3		143.3	37		82.7		82.7	71	5.72	71.1	5.70	73.3
3	6.61	151.9	6.61	151.9	38	4.31	80.0	4.30	80.0	72	4.20	81.7	4.06	83.4
4a	2.45	37.0	2.46	36.6	39a	2.11	33.3	2.12	32.8	73	4.77	64.7	4.67	67.1
4b	2.55		2.56		39b	2.22		2.24		74a	2.39	38.9	2.44	39.9
5	4.37	76.8	4.35	76.2	40	4.13	78.5	4.10	77.8	74b	2.69		2.83	
6	5.45	127.3	5.45	127.0	41		82.7		82.7	75	3.35	79.8	3.19	79.7
7	5.97	140.6	6.01	139.9	42	4.00	75.7	4.02	75.9	76	3.71	76.4	3.48	79.1
8		78.3		78.3	43a	2.22	42.5	2.20	42.5	77a	2.16	37.7	1.82	37.9
9	3.85	80.8	3.85	80.0	43b	2.64		2.43		77b	2.51		2.49	
10a	1.96	26.9	1.97	26.4	44	3.82	84.5	4.14	73.3	78	3.38	79.4	3.26	79.5
10b	2.03		2.03		45	4.13	85.6	4.24	85.0	79	3.45	80.1	3.45	80.1
11a	1.88	27.0	1.90	26.9	46a	2.11	33.7	2.14	35.2	80a	2.29	38.7	2.20	37.5
11b	2.04		2.05		46b	2.25		2.20		80b	2.56		2.55	
12	4.04	76.7	4.07	75.4	47a	1.98	30.7	1.72	27.2	81	3.22	80.7	3.28	79.7
13		81.4		81.4	47b	2.04		2.02		82	3.38	80.6	3.38	79.2
14a	2.03	44.7	2.05	43.7	48	3.09	85.6	3.17	88.5	83a	1.72	39.1	1.74	38.4
14b	2.08		2.08		49	3.21	85.3		78.3	83b	2.50		2.52	
15	3.91	73.0	3.90	72.7	50a	1.86	42.5	1.89	49.1	84	4.19	66.9	4.26	66.4
16		76.4		76.4	50b	2.50		2.44		85	3.22	83.7	3.27	83.3
17a	1.99	41.9	1.98	41.6	51	3.41	82.6	4.39	79.7	86	4.37	67.8	4.40	67.4
17b	2.24		2.22		52	3.21	83.1	3.52	83.0	87a	1.95	34.6	2.01	34.0
18	4.52	74.7	4.53	74.2	53a	1.67	32.0	2.14	39.0	87b	2.55		2.58	
19		79.6		79.6	53b	1.97		2.48		88	3.67	79.1	3.69	78.4
20	4.39	75.3	4.39	75.1	54	1.76	40.5	4.06	76.8	89	5.87	87.5	5.86	86.8
21	3.67	82.8	3.67	82.4		1.97				90	6.09	131.8	6.10	132.0
22	4.20	75.3	4.20	75.4	55		80.1		82.0	91	6.31	132.8	6.31	132.3
23a	1.89	38.5	1.90	38.0	56	3.31	83.7	4.26	77.1	92	5.13	92.3	5.01	90.7
23b	2.72		2.72		57a	2.08	32.9	2.08	34.5	93	1.58	48.9	1.49	48.4
24	3.42	81.8	3.41	81.8	57b	2.31		2.33		94	4.38	68.8	4.4	67.4
25	3.59	80.6	3.59	80.6	58	4.04	71.6	3.25	80.9	95	1.36	23.8	1.37	23.1
26a	1.73	45.9	1.74	46.13	59	3.63	72.8	3.76	71.0	96	1.74	11.8	1.74	11.4
26b	2.32		2.39		60a	4.43	70.8	1.67	41.9	97	1.40	24.1	1.47	24.3
27		76.3		76.3	60b			2.41		98	1.56	23.3	1.61	22.9
28	3.44	86.5	3.46	85.9	61	4.07	73.1	4.46	73.6	99	1.48	17.6	1.48	17.3
29a	2.02	30.2	2.02	30.2	62	4.78	74.0	4.07	82.4	100	1.49	19.8	1.50	17.6
29b	2.07		2.07		63a	2.59	35.7	4.72	77.2	101	1.45	24.9	1.47	24.3
30	3.67	76.6	3.67	76.2	63b	2.71				102	1.63	23.3	1.69	23.2
31		77.2		77.2	64	5.18	70.3	5.26	72.7	103	1.57	24.2	1.58	22.8
32a	1.89	45.6	1.91	44.8	65	5.17	72.8	4.89	73.6	104	1.39	21.9	1.40	21.4
32b	2.33		2.41		66	4.11	75.6	4.39	76.5	105	1.39	24.9	1.40	23.2
33	4.85	81.5	4.89	80.9	67	4.47	72.6	4.50	75.4	106	1.39	19.5	1.46	18.2
34		81.7		81.7	68	4.87	68.2	5.11	72.4	107	1.02	12.6	1.04	11.5
35a	2.39	48.9	2.29	48.4	69a	2.11	33.2	2.09	35.1	108			1.38	17.0
35b	2.59		2.60		69b	3.52		3.46						

Pos.: Position, δ_H_: ^1^H chemical shifts, δ_C_: ^13^C chemical shifts. ^a^^13^C chemical shifts were assigned based on HSQC and HMBC spectra.

**Table 2 toxins-13-00082-t002:** Δδ values (δ_KBTG_−δ_KBTA2_) for the CH-71 region.

Pos.	Δδ_H_	Δδ_C_	Pos.	Δδ_H_	Δδ_C_	Pos.	Δδ_H_	Δδ_C_
65	+0.08	0.0	69a	+0.10	+0.4	72	+0.15	−0.8
66	+0.05	0.0	69b	−0.30		73	−0.10	−0.3
67	0.00	0.0	70	−0.19	+1.6	74a	0.00	+1.7
68	−0.61	−0.3	71	−0.65	−6.2	74b	0.00	

## Data Availability

Not applicable. Data is provided in the manuscript and Appendix A.

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
