# Peer review of "Brevisulcenals-A1 and A2, Sulfate Esters of Brevisulcenals, Isolated from the Red Tide Dinoflagellate Karenia brevisulcata"

_toxins, 2021, doi:10.3390/toxins13020082_

Round 1
Reviewer 1 Report
This manuscript is a nice sequel to Reference 3 (Holland et al. Harmful Algae 2012 13, 47-57), now addressing minor congeners. The low yields of these congeners required working with huge (1450 L) cultures of the Karenia brevisulcata followed by doing work-up in a 200 L barrel. I personally would not have the stamina or the resources to take on such a heroic project just to get a milligram of a minor congener. I have nothing but admiration for these people.
My only negative comment about the manuscript involves the introduction. To start with, the first paragraph needs to be rewritten because there is a serious problem with plagiarism – more likely self-plagiarism.
The authors state: "The bloom also caused over 500 cases of human respiratory distress with symptoms including dry cough, severe sore throat, runny nose, skin and eye irritations, severe headaches, and facial sun-burn sensations. Respiratory distress symptoms were attributed to direct exposure to contaminated seawater and sea-spray aerosols and were similar to those caused by aerosolized brevetoxins [4,5]."
Holland (2012) had stated: "Over 500 cases of human respiratory distress were reported. Symptoms included dry cough, severe sore throat, rhinorrhoea, skin and eye irritations, severe headaches and facial sunburn sensations. The respiratory distress was attributed to direct exposure to contaminated seawater and sea-spray aerosols and resembles that caused by aerosolised brevetoxins."
I do not regard this as a serious problem having been guilty of quite a bit of self-plagiarism over the course of my career.
More serious, however, is the fact that references 4 and 5 in the manuscript are wrong. They are to structural studies by Jon Clardy and the late Koji Nakanishi with no mention of human respiratory distress or any other toxicology. Holland (2012) had cited four other references – two for each of the two sentences. I did not make the effort to check all of the references for errors but, based on the citations in Holland (2012), I believe the present authors intended 4 and 5 to be Holland's citations of Cheng (2007) and Fleming (2007). Unfortunately, those references do not actually support Holland's contention that aerosolized brevetoxin was the cause of K. brevis' respiratory distress, only that aerosolized brevetoxin-containing blooms of K. brevis were causing the distress. I am no authority on brevetoxins but a little searching around in the literature led to discovery of a paper by Janet Benson at the Lovelace Respiratory Institute and coauthored by Dan Baden (Benson et al., "Inhalation toxicology of brevetoxin 3 in rats exposed for 22 days", Environmental Health Perspectives 2005, 113, 626-631). In that paper, Benson et al. report that brevetoxin 3 produces no clinical signs of toxicity even when working with aerosol concentrations 2-3 orders of magnitude higher than the total brevetoxins measured along Florida beaches during red tides of moderate intensity. Neither Holland (2012) nor the present manuscript cite that paper. If Benson is right, the widely reported K. brevis-induced respiratory distress must be due to some other agent. Alternatively, it is just chemophobic hysteria.
Then what is the state of affairs with the brevisulcata aerosols? In the light of Benson's paper, I do not think the authors of the present manuscript should be making claims of respiratory distress by aerosolized brevisulcata toxins until appropriate testing has been carried out with purified toxins. Revision of the opening paragraph would give the authors the opportunity to straighten out the perceptions of the toxicology of brevisulcata toxins.
Author Response
Response to Reviewer 1
Manuscript ID: toxins-1074872
Title: Brevisulcenals-A1 and A2, Sulfate Esters of Brevisulcenals, Isolated from the Red Tide Dinoflagellate Karenia brevisulcata
Changes in the revised manuscript according to comments from Review 1 are highlighted in yellow.
1) According to the comments from Reviewer 1, Introduction was rewritten
Page 1, Line 44. “In addition to the devastating damages to marine animals, more than 500 patients were reported during the red tide incident. Characteristic symptoms were respiratory distress, inflammation of skin and eyes, severe headaches, and facial sun-burn sensations, which were similar to symptoms caused by Karenia brevis [4-6]. Further toxicological studies are needed to elucidate correlation between human illness and the toxins of K. brevisulcata.
2) As reviewer 1 mentioned, we have not tested toxicity of toxins from Karenia brevisulcata against humans. So the below sentence was added in Introduction
“Further toxicological studies are needed to elucidate correlation between human illness and the toxins of K. brevisulcata.”
3) Epidemiologic studies about aerosol of Karenia brevis have been conducted.
References 4 and 5 were revised and revised references 4-6 are blow.
Page 9, Line 300.
- Backer, L.C.; Fleming, L.E.; Rowan, A.; Cheng, Y.-S.; Benson, J.; Pierce, R.H.; Zaias, J.; Bean, J.; Bossart, G. D.; Jonson , D.; Quimbo, R.; Baden, D.G. Recreational exposure to aerosolized brevetoxins during Florida red tide events. Harmful Algae 2003, 2, 19-28.
- Fleming, L.E.; Kirkpatrick, B.; Backer, L.C.; Bean, J.A.; Wanner, A.; Reich, A.; Cheng, Y.S.; Pierce, R.; Naar, J.; Abraham, W.M.; Baden, D.G. Aerosolide red-tide toxins (brevetoxins) and asthma. CHEST 2007, 131, 187-194.
- Bean, J.A.; Fleming, L.E.; Kirkpatrick, B.; Backer, L.C.; Nierenberg, K.; Reich, A.; Cheng, Y.S.; Wanner, A.; Benson, J.; Naar, J.; Pierce, R.; Abraham, W.M.; Kirkpatrick, G.; Hollenbeck, J.; Zaias, J.; Mendes, E.; Baden, D.G. Florida red tide toxins (brevetoxins) and longitudinal respiratory effects in asthmatics. Harmful Algae 2011, 10, 744-748.
4) Reference numbers 7-22 are changed.

Reviewer 2 Report
Manuscript deal with isolation and structural investigation of two polycyclic ether toxins: brevisulcenal-A1 and brevisulcenal-A2. I have no serious reservations about the quality of the work.It is very good work, however, some minor points should be improved:
1. Liquid chromatograph used for separation should be described.
2. Parameters of diol-column should be provided.
3. Page 3 line 98 "The UV spectrum of KBT-A2 was collected at 227 nm ". Spectrum cannot be registered at single wavelength. Similar at page 6 line 149
4. Please edit the Table 1 to show continuity of columns. It is not very clear.
5. Fig. S1, S4, S8 and S9 have to low resolution.
6. In some point the UV spectra are mention, but they are not presented nor discussed.
Author Response
Response to Reviewer 2
Manuscript ID: toxins-1074872
Title: Brevisulcenals-A1 and A2, Sulfate Esters of Brevisulcenals, Isolated from the Red Tide Dinoflagellate Karenia brevisulcata
Changes in the revised manuscript according to comments from Review 2 are highlighted in pink.
- Liquid chromatograph used for separation should be described.
According to the comments from the review 2, the liquid chromatograph was described in Page 7, Line 202.
“A JASCO PU-98 pump and a JASCO UV-970 detector were used for liquid chromatography.”
- Parameters of diol-column should be provided.
According to the comments from the review 2, the parameter of the diol column was described in Page 8, Line 232.
“Brevisulcenals were isolated from the neutral fractions of 1450 L of bulk cultures by column chromatography using a diol cartridge (500 mg)”
- Page 3 line 98 "The UV spectrum of KBT-A2 was collected at 227 nm ". Spectrum cannot be registered at single wavelength. Similar at page 6 line 149
According to the comments from the review 2, the sentence was revised.
Page 3, Line 109. “The UV maximum of KBT-A2 was observed at 227 nm (e 1.2 × 104).”
Page 6, Line 163. “The UV absorption of brevisulcenal-A1 (KBT-A1) was at 225 nm (e 0.6 × 104) indicating that KBT-A1 had an enal side chain.”
- Please edit the Table 1 to show continuity of columns. It is not very clear.
Table 1 was edited. Compounds and position were inserted at the top cells in the second page..
- Fig. S1, S4, S8 and S9 have to low resolution.
Figures S1, S4, S8, and S9 were replaced with high resolution spectra.
- In some point the UV spectra are mention, but they are not presented nor discussed.
According to the comments from the review 2, UV spectra of KBT-A1 and KBT-A2 were discussed.
Page 3, Line 109. “The UV absorption suggested that KBT-A2 has an enal side chain analogous with KBT-G.”
Page 7, Line 163. “The UV absorption of brevisulcenal-A1 (KBT-A1) was at 225 nm (e 0.6 × 104) indicating that KBT-A1 had an enal side chain.”

Round 2
Reviewer 1 Report
I recommend publication of the revised manuscript.
However, I am disappointed that the authors did not want to cite the paper by Janet Benson which showed little pulmonary toxicity of purified brevetoxin at concentrations and duration of exposure to animals much higher than environmental exposure of humans. The conclusion I draw from Benson's paper is that the respiratory distress results from some as yet uncharacterized K. brevis toxin. The same may be true for K. brevisulcata as well. The situation with Karenia may parallel that of Alexandrium where there there are well-characterized endotoxins - saxitoxins et al. - along with lytic exotoxins for which structural characterization has thus far eluded investigators.